# Comparative Bioavailability Study of Jaspine B: Impact of Nanoliposomal Drug Delivery System on Pharmacokinetics

**DOI:** 10.3390/pharmaceutics17070807

**Published:** 2025-06-22

**Authors:** Biwash Ghimire, Pradeep Giri, Sameena Mateen, Srinath Pashikanti, Ali Aghazadeh-Habashi

**Affiliations:** College of Pharmacy, Idaho State University, Pocatello, ID 83209, USA; biwashghimire@isu.edu (B.G.); pradeepgiri@isu.edu (P.G.); sameenamateen@isu.edu (S.M.); srinathpashikanti@isu.edu (S.P.)

**Keywords:** Jaspine B, pachastrissamine, liposomes, drug delivery, analytical method validation, pharmacokinetics

## Abstract

**Background/Objectives**: Jaspine B, a synthetic analog of anhydrophytosphingosine, demonstrates significant anticancer activity; however, its clinical application is hindered by its poor oral bioavailability, resulting in suboptimal systemic exposure. This study aimed to enhance the pharmacokinetic properties of Jaspine B by developing a liposomal delivery system. **Methods**: Jaspine B-loaded liposomes were formulated using a microfluidic approach and characterized by transmission electron microscopy (TEM) to assess particle morphology and size distribution. A sensitive and selective LC-MS/MS assay was developed and fully validated to quantify Jaspine B in rat plasma. The assay revealed excellent linearity across a broad concentration range and high intra- and inter-day precision. A pharmacokinetic study was conducted in Sprague Dawley rats to evaluate the influence of liposomal encapsulation on the pharmacokinetic profile of Jaspine B. **Results**: The liposomal formulation accelerated the absorption of Jaspine B, reaching the maximum concentration (T_max_) at 2 h as opposed to 6 h in plain Jaspine B. The half-life (t_1/2_) increased significantly from 7.9 ± 2.3 h to 26.7 ± 7.3 h. The area under the curve (AUC_0–∞_) increased over two-fold from 56.8 ± 12.3 ng.h/mL to 139.7 ± 27.2 ng.h/mL, suggesting increased systemic drug exposure. Similarly, the drug molecule’s mean residence time (MRT) increased over three-fold. **Conclusions**: These results indicate that liposomal formulation enhances the pharmacokinetics of Jaspine B, prolonging its body circulation and exposure, which explains the improved therapeutic outcomes we observed in our previous pharmacodynamic study.

## 1. Introduction

Jaspine B is a naturally occurring cyclic anhyrophytosphingosine originally derived from marine sponges Pachastrissamine sps. [1] and Jaspis sps. [2]. Initially isolated in 2002, Jaspine B has gained significant research interest due to its potent anticancer properties [3], with both Jaspine B and its analogs exhibiting cytotoxic activity across a broad spectrum of cancer cell lines in vitro [4,5,6].

Jaspine B exerts its anticancer effect through multiple mechanisms, primarily by disrupting sphingolipid pathways, leading to both apoptotic and non-apoptotic cell death [4,7]. Jaspine B acts as an inhibitor of sphingomyelin synthase, thereby promoting the intracellular accumulation of ceramide and other bioactive sphingolipid metabolites [8]. In addition to inhibiting sphingomyelin synthase, Jaspine B also inactivates sphingosine kinase 1 and 8 and suppresses Forkhead box O3 (FOXO3) activity, collectively contributing to the induction of apoptosis [9].

Despite its promising therapeutic potential, the clinical translation of Jaspine B is hindered by its unfavorable pharmacokinetic properties, most notably its poor oral bioavailability. Preclinical studies in rats have reported an oral bioavailability as low as 6.2%, which is primarily attributed to its low aqueous solubility, extensive first-pass metabolism, and rapid systemic clearance [10]. Zhang et al. investigated various chemical modifications to generate Jaspine B analogs, aiming to enhance their anticancer activity [11]. Choi et al. employed the taurocholate supplementation method to improve bioavailability. The results show a significant improvement in Jaspine B’s pharmacokinetic profile with taurocholate supplementation, resulting in a more than six-fold increase in bioavailability [10]. This pharmacokinetic improvement, however, was not evaluated for its correlation with pharmacodynamic effect.

Liposomes have emerged as promising carriers for enhancing the oral bioavailability of poorly soluble drugs, such as Jaspine B, by improving solubility, stability, and absorption in the gastrointestinal (GI) tract. Encapsulation within liposomes can protect medicines from harsh GI conditions and enzymatic degradation while promoting lymphatic transport and facilitating transcellular absorption across intestinal epithelia [12,13,14]. Moreover, surface modifications and the use of bile salts or PEGylated liposomes can further enhance mucosal permeability and prolong GI residence time, thereby increasing drug exposure and bioavailability [13,15,16,17].

As a novel approach to enhancing the efficacy of Jaspine B in vitro and in vivo, we developed a nano-liposomal formulation and conducted a pharmacodynamic study, in which the liposomal formulation was found to improve the therapeutic efficacy of Jaspine B in tumor-bearing mice [6]. We assume that the improvement in the pharmacodynamic effects of liposomal Jaspine B is attributed to the enhanced pharmacokinetic properties resulting from liposomal drug delivery. This observation aligns with previous reports that highlight liposomal formulations as promising drug delivery systems for orally administered water-insoluble drugs, resulting in a several-fold increase in bioavailability [15,18]. This study aims to validate this assumption by determining the pharmacokinetic parameters of Jaspine B following oral administration at a 5 mg/kg dose, consistent with that used in our prior pharmacodynamic investigation [6].

Our liposomal formulation was composed primarily of 1,2-distearoyl-sn-glycero-3-phosphocholine (DSPC) as a structural lipid. Incorporating cholesterol into the liposomal structure was necessary to modulate the phospholipid bilayer’s fluidity, control membrane permeability and maintain overall stability [19]. Polyethylene glycol (PEGylation) was added to liposomes using PEG-containing phosphoethanolamine (DSPE-PEG2000-COOH). The PEGylation of liposomes is very important when addressing certain limitations of conventional liposomes. PEGylated liposomes have a drug-loading capacity that is improved by 90%, resulting in an efficient formulation [16]. Surface modification with PEG also improves gastrointestinal tract stability and therapeutic outcomes [20,21].

While other formulations, such as solid lipid nanoparticles (LNPs) and micelles, are also viable strategies for drug delivery, they encapsulate hydrophobic molecules poorly. Jaspine B, a sphingolipid, does not encapsulate well in the core of reverse micelles produced using the ionic lipids in the LNPs [22]. Micelles are also suitable drug carriers for entrapping hydrophobic molecules; however, they are relatively unstable when exposed to environmental changes. When micelles are diluted below their critical micellar concentration (CMC), they dissociate, leading to drug leakage [23]. Liposomes provide a comprehensive balance between drug loading and stability for encapsulating hydrophobic molecules and were therefore chosen as drug carriers in this study.

## 2. Materials and Methods

### 2.1. Reagents

The in-house synthesis and scale-up of Jaspine B was carried out based on the previously published method [24]. Cholesterol, 1,2-distearoyl-sn-glycero-3-phosphocholine (DSPC), and 1,2-distearoyl-sn-glycero-3-phosphoethanolamine-N- [amino (polyethylene glycol)-2000] carboxylic acid (DSPE-PEG2000-COOH) were purchased from Avanti Polar Lipids (Alabaster, AL, USA). The water, acetonitrile, methanol, and formic acid used in the analysis were of LC-MS grade and purchased from Fisher Scientific (Fair Lawn, NJ, USA).

### 2.2. Preparation and Characterization of Jaspine B Liposomal Formulation

Liposomes were formulated using a microfluidic method on the NanoAssemblr^TM^ platform (Precision Nanosystems, Vancouver, BC, Canada). The detailed formulation, characterization, and stability testing process are described elsewhere [6]. Briefly, a mixture of lipids composed of cholesterol, DSPC, and DSPE-PEG2000-COOH in a molar ratio of 2:4:4 was dissolved in ethanol at 10 mg/mL. Jaspine B was also dissolved in ethanol separately at a concentration of 2 mg/mL. These two ethanolic solutions were mixed at equal volumes to achieve a final concentration of lipids of 5 mg/mL and Jaspine B of 1 mg/mL, respectively, with the ethanol ratio kept constant. Liposomal formulations were produced by simultaneous injection of the ethanolic solution and phosphate-buffered saline (PBS) into the NanoAssemblr cartridge at a flow rate ratio (FRR) of 2:1 and total flow rate (TFR) of 8 mL/min. Since the parameters FRR and TFR significantly affect particle size, they were kept constant throughout the formulation [25]. The final solution, consisting of an aqueous-to-ethanolic ratio of 2:1, was collected and dialyzed against 10 mM PBS at room temperature, followed by a desalting step using 10 kDa molecular weight cutoff filters.

The morphology and liposome size were analyzed using a transmission electron microscope (TEM) (Zeiss EM900, Carl Zeiss NTS Ltd., Oberkochen, Baden-Württemberg, Germany). Briefly, 20 µL of the sample was deposited onto mesh carbon-coated copper grids and allowed to stand for 15 min to develop a thin film. The excess solvent was drained using blotting paper, and the remaining solvent was allowed to air-dry. The grids were visualized under a transmission electron microscope (TEM) and processed using Gatan microscopy suite software V3.6 (Gatan Inc., Pleasanton, CA, USA). One hundred data points were generated from the microscopy data, and the Gaussian distribution was calculated using Origin Visual basic software V10.0 (Origin Lab Corporation, Northampton, MA, USA). For the zeta potential measurements, 100 µL of sample was diluted with 900 µL of phosphate-buffered saline (PBS) and added to folded capillary cells in duplicates. Five measurements were carried out for each replicate in BeNano 180 zeta pro (Bettersize Instruments, Liaoning, China).

Entrapment efficiency (%EE) was determined using a previously developed methodology with small modifications [6,26]; Here, 100 µL aliquots of previously dialyzed liposome suspension were mixed with 100 µL of triton-X100 and mixed well. The mixture was centrifuged at 20,000× *g* for 30 min; Then, 20 µL of internal standard at 200 ng/mL was added to each sample and injected into the LC-MS/MS system. A standard calibration curve was used to calculate the concentration of Jaspine B present in the solution. The entrapment efficiency was calculated using the following equation:$$EE\left(\%\right)=\frac{Total\;mass\;of\;Jaspine\;B\;loaded\;into\;the\;liposome}{Total\;mass\;of\;Jaspine\;B\;used\;for\;the\;formulation}$$

### 2.3. LC-MS/MS System

The LC-MS/MS system consisted of a Shimadzu liquid chromatography platform (Columbia, MD, USA) equipped with a binary pump (LC-30AD), autosampler (SIL-30AC), system controller (CBM-20A), degasser (DGU-20A5R), and column oven (CTO-20A), coupled to an AB Sciex QTRAP 5500 mass spectrometer (SCIEX, Foster City, CA, USA) with an electrospray ionization (ESI) source. Chromatographic data acquisition was performed using Analyst 1.7 software. Quantitative analysis was conducted with MultiQuant 3.0 (SCIEX, Foster City, CA, USA). The separation of analytes was achieved on a Synergi™ Fusion-RP column (2.5 µm, 100 × 2 mm; Phenomenex, Torrance, CA, USA) in positive ion mode.

The mobile phase consisted of 0.1% formic acid in water (solvent A) and acetonitrile (solvent B), delivered at a flow rate of 0.2 mL/min via gradient elution. The gradient was initiated at 10% B, ramped to 25% over 2 min, then sequentially increased to 50%, 75%, and 90% over the next 2 min, and maintained at 90% for an additional 7 min. Detection of Jaspine B and internal standards was performed in positive ion mode using multiple reaction monitoring (MRM).

### 2.4. Preparation of Working Solutions and Calibration Standards

A stock solution of Jaspine B (1 mg/mL) was prepared by dissolving the compound in methanol. Working solutions at concentrations of 0.5, 1, 2, 4, 8, and 16 ng/mL were prepared to construct a standard calibration curve of Jaspine B by serially diluting the stock solution (1 mg/mL) with methanol. Similarly, a 5 ng/mL working solution of Spisulosine, the internal standard (IS), was prepared by diluting the stock solution of 1 mg/mL Spisulosine in methanol. Calibration standard solutions were prepared by spiking blank rat plasma with 100 µL of Jaspine B and IS working solutions.

### 2.5. Sample Preparations

An aliquot of 100 µL rat plasma was combined with 100 µL of a 5 ng/mL IS solution and mixed thoroughly. Subsequently, 1 mL of cold methanol (−20 °C) was added, followed by vortexing for 30 s. An additional 500 µL of cold acetonitrile (−20 °C) was then added, and the mixture was vortexed to ensure thorough mixing. The mixture was then placed in a water bath at 60 °C for 10 min to allow for the complete dissolution of the analyte in the solution. After removing the mixture from the water bath, it was shaken well and centrifuged for 20 min at 17,000× *g*. The resulting supernatant was transferred to a glass test tube and dried under a gentle stream of nitrogen gas. The dried supernatant was reconstituted in 100 µL of methanol, and an aliquot of 10 µL was injected into the LC-MS/MS system. The analytical method validation data are presented as descriptive statistics (average recovery (%) ± SD), and the coefficient of variance (CV < 10%) is considered a metric for precision.

### 2.6. Pharmacokinetic Study

The pharmacokinetic study was conducted in two groups of male Sprague Dawley rats (N = 3 per group; 8 weeks old; 250–300 g), administered either a Jaspine B suspension (formulated in DMSO:PEG2000:PBS at a ratio of 2:4:4) or a liposomal formulation of Jaspine B. Animals were procured from Charles River Laboratories (Hollister, CA, USA). Following a 3-day acclimatization period, the rats were surgically cannulated in the right jugular vein under anesthesia and allowed to recover overnight. Each group received an oral dose of 5 mg/kg Jaspine B via gavage. The Jaspine B suspension was freshly prepared immediately prior to administration. This dose was chosen to be consistent with our previous pharmacodynamic study [6]. Although, based on our last analysis, the liposomal formulation was stable for more than two months in PBS at pH 7.4 and 4 °C, the liposomal formulation was still prepared a day before the experiment to avoid any possible degradation due to storage.

Serial blood samples were collected via the jugular vein cannula at 15, 30, 60, 120, 360, and 720 min post-dose. Following each collection, the cannula was flushed with normal saline and subsequently filled with a heparinized saline lock solution. At 24 h post-dose, the rats were euthanized using CO_2_, and terminal blood samples were obtained by cardiac puncture. Pharmacokinetic parameters were analyzed using PKSolver software add-in for Microsoft Office V15 employing non-compartmental analysis [27].

### 2.7. Statistical Analysis

Statistical analyses were performed using GraphPad Prism 8.0 statistical software (San Diego, CA, USA), and the results are expressed as the mean ± SD. Comparison between groups was carried out using Student’s *t*-test, with *p* < 0.05 considered statistically significant.

## 3. Results

### 3.1. Liposome Formulation and Characterization

The Jaspine B liposomal formulation was analyzed for encapsulation efficiency (EE%) and size determination. The detailed results regarding size optimization are reported in our previous study [6]. We chose the formulation parameters of FRR = 2:1 and TFR = 8 mL/min due to their high entrapment efficiency. The TEM results show that the mean particle size of spherical liposomes was 127.5 ± 61.2 nm (Figure 1). The entrapment efficiency (%EE) was calculated to be 97% [6,26], and the mean zeta potential of the liposomes was −5.60 ± 0.33 mV. Jaspine B liposomes remained stable for more than two months in PBS (pH 7.4) at 4 °C, with no significant changes in their size or EE%. However, for animal studies, the fresh liposomes were prepared and analyzed before dosing to avoid any possible degradation.

### 3.2. LC-MS/MS Analysis of Jaspine B in Rat Plasma

We developed and validated an analytical method to measure the concentration of Jaspine B in rat plasma using multiple reaction monitoring (MRM) modes in an LC-MS/MS system. The MS was optimized in positive ion mode after the direct injection of Jaspine B and IS solutions at a concentration of 100 ng/mL (Table 1, Figure 2 and Figure 3).

### 3.3. Analytical Method Validation

#### 3.3.1. Specificity

The specificity of the method was confirmed using five different rat blank plasma samples and rat plasma samples spiked with 1 ng/mL of Jaspine B and 5 ng/mL of the IS (Figure 3).

#### 3.3.2. Accuracy and Precision

A set of six concentrations of Jaspine B in rat plasma was analyzed to create calibration curves ranging from 0.5 ng/mL to 16 ng/mL. The results of intra-day and inter-day assays are shown in Table 2. The intra-day assays showed an average recovery of analytes ranging from 92.23% to 108.28%, with the variance ranging from 0.77% to 8.76%. For inter- day assays, the method’s average accuracy ranged from 99.23% to 103.64%, with variance ranging from 0.78% to 7.88%.

### 3.4. Pharmacokinetics

The Jaspine B plasma concentration of rats that received an oral administration of Jaspine B suspension reached the maximum value (C_max_) of 4.59 ± 1.12 ng/mL at 6.00 ± 0.02 h (T_max_). Although there was no significant difference in the C_max_ after administration of the liposomal formulation, its T_max_ was 2.00 ± 0.02 h, which was achieved significantly faster than the administration of the drug alone (*p* > 0.0001) (Figure 4).

Jaspine B’s half-life (t_1/2_) was calculated as 7.89 ± 2.34 h, slightly longer than the previously reported value of 5.5 ± 1.1 h [10]. The liposomal formulation increased the t_1/2_ more than three-fold (26.67 ± 7.32 h, *p* = 0.0370) (Table 3).

The liposomal formulation of Jaspine B significantly impacted its body exposure, as indicated by the area under the curve (AUC_0-t_) and mean residence time (MRT). The AUC after 24 h increased from 47.96 ± 12.81 ng.h/mL to 88.20 ± 3.85 ng.h/mL (*p* = 0.0303) and AUC_0–∞_ increased more than two-fold (*p* = 0.0244). The MRT of Jaspine B increased from 12.86 ± 3.65 to 39.13 ± 9.70 h (*p* = 0.0202), indicating significantly longer circulation time after administration of the liposomal formulation compared with the plain Jaspine B formulation (Table 3). This observation may explain the enhanced therapeutic outcome, which concurs with our previous study’s improved pharmacodynamic effect of Jaspine B liposomes [6].

## 4. Discussion

The poor bioavailability of Jaspine B presents a significant barrier to its effective therapeutic and anticancer application. The pharmacokinetic study demonstrated that a liposomal formulation of Jaspine B markedly enhanced its bioavailability relative to the conventional oral suspension. The formulation resulted in a prolonged half-life, indicative of sustained systemic exposure, and a significantly increased AUC, reflecting improved overall drug absorption. These findings underscore the potential of liposomal delivery systems to overcome Jaspine B’s pharmacokinetic limitations and support its use in optimizing clinical efficacy. These results complement the findings from our laboratory’s report, which demonstrated that liposomal Jaspine B exhibited enhanced therapeutic efficacy in both in vitro and in vivo studies [6]. A cell viability study conducted in Yamato cell lines showed that the IC50 of plain Jaspine B and liposomal Jaspine B was 0.36 ± 0.07 µM and 0.06 ± 0.01 µM, respectively, suggesting improved in vitro efficacy. This pharmacodynamic study demonstrated that liposomal Jaspine B outperformed plain Jaspine B in reducing tumor size in mice with synovial sarcoma. We assumed this observation could be due to the improved pharmacokinetics resulting from the oral administration of the Jaspine B liposomal formulation. Therefore, the current study was designed to validate this assumption through Jaspine B pharmacokinetic studies.

The developed and validated LC-MS/MS analytical method for quantifying Jaspine B was sensitive, specific, and robust. Sample preparation employed a protein precipitation technique, which established a reliable procedure with a lower limit of quantification of 0.5 ng/mL. This method was more sensitive than the methods developed in previous studies, which were limited by their higher quantification limit (25 ng/mL) [10,28]. Additionally, the internal standard (spisulosine) used in our study had a similar molecular structure to the analyte, resulting in improved accuracy, linearity, and precision.

The development of a liposomal Jaspine B formulation represents a significant advancement in overcoming the pharmacokinetic limitations of this promising therapeutic agent. Our findings indicate that the liposomal formulation exhibits a superior pharmacokinetic profile relative to the conventional Jaspine B formulation.

The significantly shorter time to reach maximum concentration (T_max_) observed with the liposomal formulation (2.00 ± 0.0 h) compared to the Jaspine B suspension (6.00 ± 0.00 h) indicates enhanced absorption kinetics when the drug is encapsulated in liposomes (Figure 4). This effect can be attributed to the improved solubility of Jaspine B achieved through liposomal encapsulation. Our results agree with previous studies demonstrating that liposomal delivery systems reduce absorption time, thereby lowering T_max_ [29,30,31]. Although no significant difference was observed in the maximum concentration (C_max_) of Jaspine B between the two groups, the liposomal formulation prolonged the half-life of Jaspine B by over three-fold, increasing it from 7.89 ± 2.34 h to 26.67 ± 7.32 h (Table 3). The observed shorter T_max_ and prolonged elimination half-life (t_1/2_) following liposomal administration of Jaspine b can be attributed to the pharmacokinetic-modulating properties of the liposomal carrier. As one possible explanation, the liposomal formulation likely enhances absorption by improving the solubility and stability of the drug. Simultaneously, the encapsulated drug is protected from enzymatic degradation and systemic clearance, possibly allowing for prolonged circulation, which could explain the observed longer apparent t_1/2_. In addition, a similar increase in MRT could clarify that the formulation spends more time in circulation than the drug itself, making it available for a longer duration of action. Although we did not directly study the mechanism of this prolonged circulation, our findings align with numerous studies that have established that liposomal formulations enhance the t_1/2_ and residence time in the blood [32,33,34]. This property may be attributed to the ability of PEGylated liposomes to repel plasma components, such as opsonins, thereby avoiding uptake by reticuloendothelial cells [35,36]. Consequently, this reduces the metabolism of the liposomes, allowing them to remain in circulation for extended periods.

The significant increase in AUC, as a measure of total drug exposure, may also contribute to the enhanced therapeutic efficacy of the liposomal formulation observed in our previous study [6]. Collectively, these effects indicate enhanced delivery efficiency and a depot-like pharmacokinetic profile of the liposomal system. The integration of accelerated absorption, extended systemic circulation, and increased overall drug exposure positions the liposomal formulation as a promising strategy to improve the therapeutic efficacy of drugs with suboptimal pharmacokinetic characteristics. The added advantage that liposomal formulations offer may be extrapolated to their potential for utilizing lower and less frequent dosing of the cytotoxic effects, thereby reducing adverse effects. These drugs may achieve increased tumor availability owing to their prolonged systemic residence time and the passive accumulation of liposomes facilitated by the enhanced permeability and retention (EPR) effect [37]. Under these conditions, the anticancer efficacy of the drug is expected to improve, while the development of resistance to therapy may be diminished.

Studies on the clinical implications of Jaspine B have been limited to various types of cancers, including gastric cancer and synovial sarcoma [6,7]. However, inhibiting ceramide synthase and sphingosine kinase enzymes also unlocks the possibility of using Jaspine B to treat inflammatory and immune-related diseases, including rheumatoid arthritis, inflammatory bowel disease, and Alzheimer’s disease [38,39]. In a previous study, Jaspine B was shown to have high distribution in the brain, kidney, and heart [10]. Liposomal formulations can be useful in reducing systemic toxicity and enhancing therapeutic efficacy by utilizing passive targeting through enhanced permeability and retention (EPR) effects [40]. This phenomenon is specifically important in the case of cytotoxic drugs such as Jaspine B that can interact with numerous enzymes, producing adverse drug interactions.

Despite the improvements shown by liposomal formulations in the pharmacokinetics of Jaspine B, this study has some limitations. It was designed as a pilot study to assess the feasibility, establish systemic exposure parameters, and generate preliminary pharmacokinetic data to explain the observed enhanced anticancer effects of Jaspine B after liposomal drug delivery [6]. This study lacks in vitro release profiles, simulated gastric and intestinal fluid stability data, and in vivo tissue distribution and cytotoxicity data. Such investigations are planned for future studies. The minimal sample size was chosen to limit animal use while still enabling the estimation of key parameters. A larger study with higher animal numbers will follow, as systemic behavior is well-characterized in this context.

Although statistical power is limited with N = 3 and inferential analyses were not performed, the data provide valuable insights into drug exposure profiles and serve as the foundation for future confirmatory studies with adequate power and larger sample sizes. Moreover, the observed pharmacokinetic trends are consistent with our observation in the pharmacodynamic study [6], further supporting the reliability of the data despite the small sample size.

## 5. Conclusions

In this study, a sensitive, selective, and robust LC-MS/MS method was developed and validated for the quantification of Jaspine B in liposomal formulations and rat plasma. This method was subsequently applied to investigate the comparative pharmacokinetics of the liposomal formulation of Jaspine B in rats.

Our results demonstrate that liposomal encapsulation significantly enhances the pharmacokinetic parameters of Jaspine B by reducing the time required to reach maximum plasma concentration, prolonging the half-life, increasing oral bioavailability, total body exposure, and circulation time, and ultimately leading to improved therapeutic outcomes. This observation is consistent with the results of our pharmacodynamic study [6]. However, due to the study being a pilot project, it has some shortcomings, including a small sample size and a lack of biodistribution data. These limitations will be addressed in a future study with a larger sample size.

## Figures and Tables

**Figure 1 pharmaceutics-17-00807-f001:**
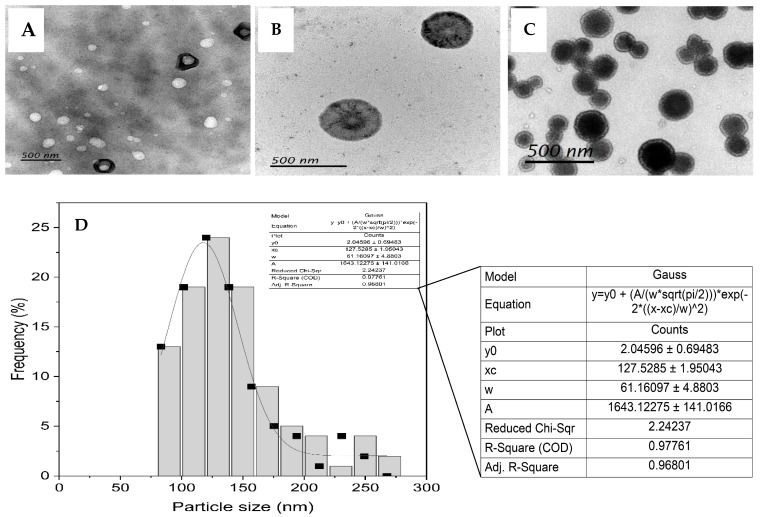
Variable magnification TEM images of Jaspine B liposomes formulated using a microfluidic method with FRR = 2:1 and TFR = 8 mL/min (**A**–**C**). The Gaussian distribution of the size of Jaspine B-entrapped liposomes (**D**). The diameter of 100 particles from different TEM images was randomly analyzed to generate a Gaussian frequency–particle size plot. The mean of the Gaussian distribution (xc) corresponds to the average size of the liposomal formulation. All of the size bars represent 500 nm in length. FRR: flow rate ratio; TFR: total flow rate; TEM: transmission electron microscopy.

**Figure 2 pharmaceutics-17-00807-f002:**
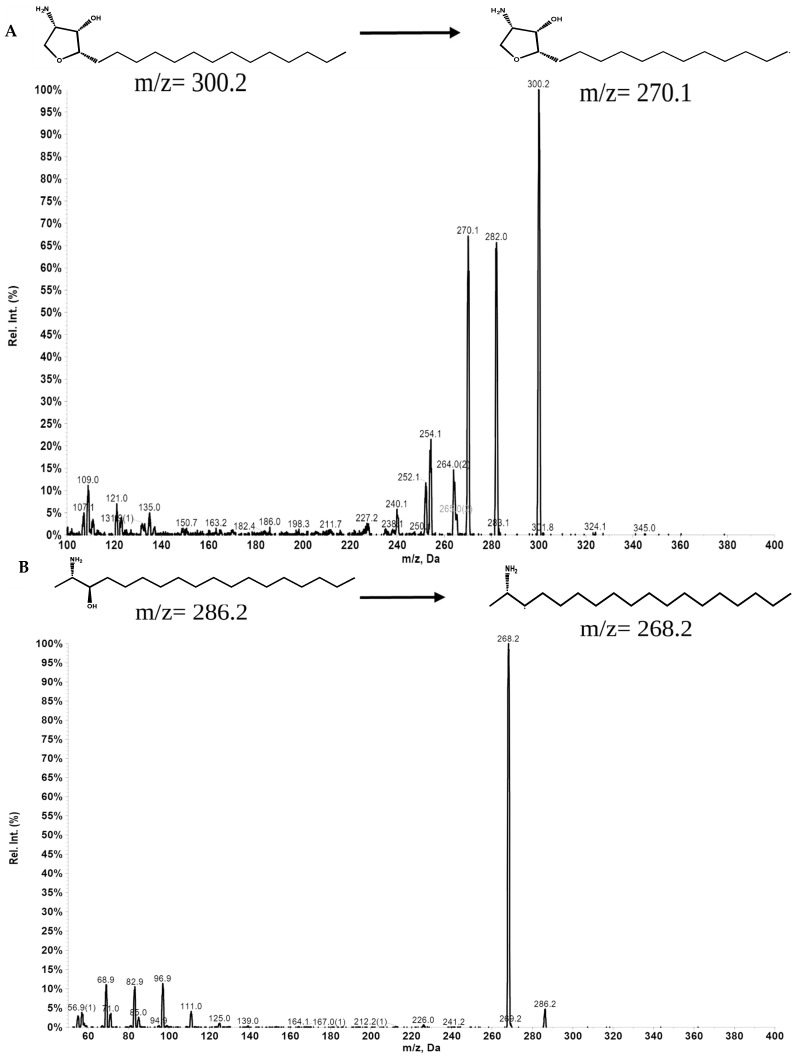
(**A**) Product ion scan and multiple reaction monitoring (MRM) chromatogram of Jaspine B. (**B**) Product ion scan and multiple reaction monitoring (MRM) chromatogram of Spisulosine (IS).

**Figure 3 pharmaceutics-17-00807-f003:**
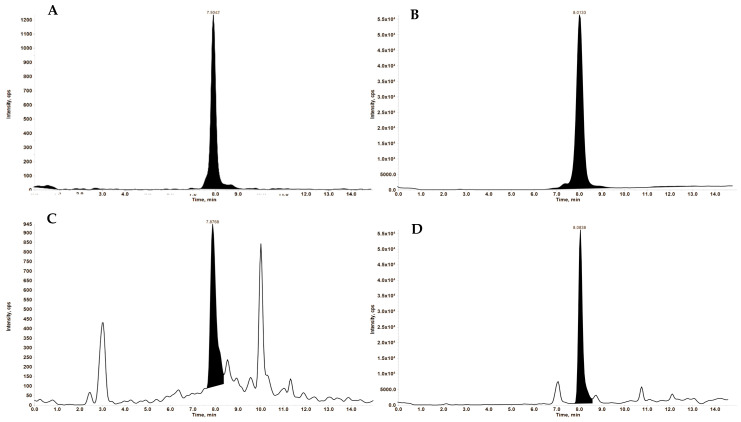
Representative extracted ion chromatograms (XIC). (**A**) Jaspine B at 1 ng/mL in Methanolic solution. (**B**) Spisulosine (IS) at 5 ng/mL in Methanolic solution. (**C**) Jaspine B at 1 ng/mL in plasma matrix. (**D**) of Spisulosine at 5 ng/mL in the plasma matrix. MRM transitions of 300.2→270.1 and 286.2→268.2 were used to generate the above-extracted Jaspine B and IS ion chromatograms, respectively.

**Figure 4 pharmaceutics-17-00807-f004:**
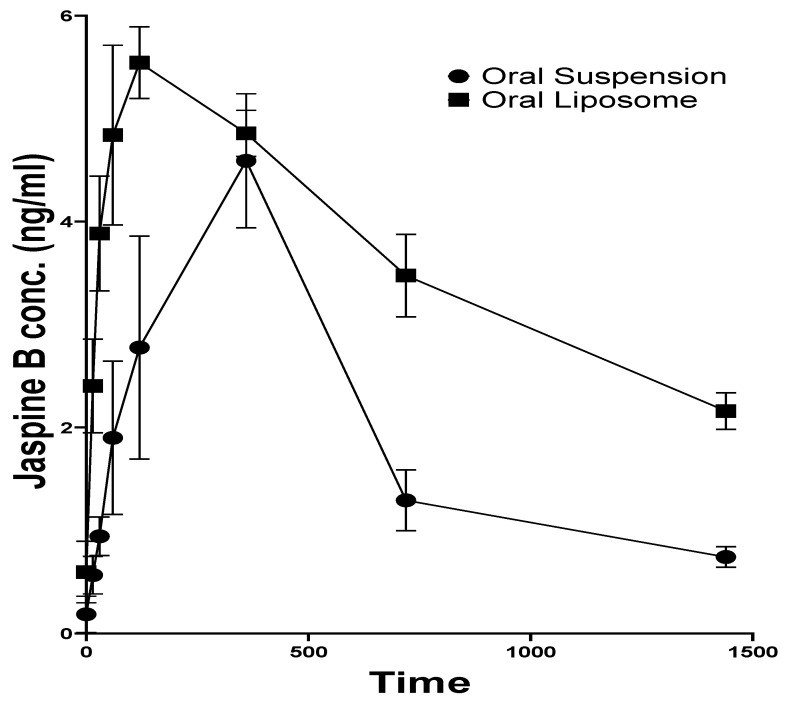
Plasma concentration–time profile of Jaspine B in rats after oral administration of Jaspine B suspension (
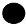
) or liposomal formulation (
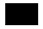
) (N = 3). Pharmacokinetic analysis was performed using PKsolver, and comparisons between groups were made using Student’s *t*-test. Statistical significance was set at *p* < 0.05.

**Table 1 pharmaceutics-17-00807-t001:** LC-MS/MS ion acquisition parameters (MRM mode) for identification and confirmation of Jaspine B and Spisulosine.

Compound	Q1 Mass	Q3 Mass	DP (Volts)	EP (Volts)	CE (Volts)	CXP (Volts)
Jaspine B	300.2	270.1	100	10	30	15
	300.2	282.0	100	10	30	15
	300.2	254.2	100	10	30	15
Spisulosine	286.2	268.2	50	5	30	15
	286.2	68.9	50	5	30	15
	286.2	56.9	50	5	30	15

**Table 2 pharmaceutics-17-00807-t002:** Intra- and inter-day accuracy and precision of Jaspine B analysis.

Conc (ng/mL)	Intra-Day Precision		Inter-Day Precision	
	Average Recovery (%) ± SD	CV (%)	Average Recovery (%) ± SD	CV (%)
0.5	108.28 ± 7.31	6.75	99.23 ± 7.82	7.88
1	92.25 ± 8.08	8.76	100.24 ± 5.81	5.80
2	96.88 ± 3.85	3.98	102.64 ± 5.65	5.51
4	97.69 ± 7.27	7.44	99.81 ± 0.78	0.78
8	100.87 ± 3.97	3.93	100.15 ± 5.08	5.07
16	100.07 ± 0.77	0.77	99.82 ± 1.09	1.10

Data are presented as average (%) ± SD, and CV < 10% was considered acceptable precision.

**Table 3 pharmaceutics-17-00807-t003:** PK parameters after an oral dose of 5 mg/kg Jaspine B suspension (JB-S) or liposomal formulation (JB-L).

Formulation	T_max_ (h)	C_max_ (ng/mL)	t_1/2_ (h)	AUC_0-t_ (ng.h/mL)	AUC_0–∞_ (ng.h/mL)	MRT (h)
JB-S	6.00 ± 0.02	4.59 ± 1.12	7.89 ± 2.34	47.96 ± 12.81	56.77 ± 12.30	12.86 ± 3.65
JB-L	2.00 ± 0.02	5.54 ± 0.61	26.67 ± 7.32	88.20 ± 3.85	139.69 ± 27.21	39.13 ± 9.70
*p* value	<0.0001	0.2840	0.0370	0.0303	0.0244	0.0202

The values are presented as mean ± SD. *p* < 0.05 considered statistically significant.

## Data Availability

The data presented in this study are available on request from the corresponding author.

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
