# Peer review of "Comparative Bioavailability Study of Jaspine B: Impact of Nanoliposomal Drug Delivery System on Pharmacokinetics"

_pharmaceutics, 2025, doi:10.3390/pharmaceutics17070807_

Round 1

Reviewer 1 Report (Previous Reviewer 1)

Comments and Suggestions for Authors

The authors should provide the detailed characterizations for the detailed formulations, characterization, and stability.

The concentration of lipids and the drug in the ethanol solution is noted as 10 mg, and Jasbine B is 2mg. However, further clarification is needed on the exact ratio of ethanol in the solution.

Additionally, the authors should provide more details about the working solution.

It would be helpful to see why the authors added the Jasmin dose as 5 m/kg.

The details of these abbreviation, TFF and TRF

The abbreviations TFF and TRF need to be fully explained, and more discussion is required, with additional supporting references."

More discussion and supported references are required.

What about the degradation of liposomes after oral application.

Author Response

Reviewer 1:

We appreciate your careful evaluation of our manuscript. Please find our detailed responses below. Revisions have been made accordingly and are indicated in the updated manuscript as highlighted text in relevant sections.

Comments 1: The authors should provide the detailed characterizations for the detailed formulations, characterization, and stability.

Response 1: Thank you for pointing this out. We agree with this comment. The requested information is revised and provided under section 2.2. Preparation and Characterization of Liposomal Formulation of Jaspine B…, line 99, and under result section 3.1. Liposome formulation and characterization, line 187 and Figure 1.

Comments 2: The concentration of lipids and the drug in the ethanol solution is noted as 10 mg, and Jasbine B is 2mg. However, further clarification is needed on the exact ratio of ethanol in the solution.

Response 2: Thank you for your comment. We explained it under section 2.2. Preparation and Characterization of Liposomal Formulation of Jaspine B…, line 99 as follows.  

Jaspine B was also dissolved in ethanol separately at a concentration of 2 mg/mL. These two ethanolic solutions were mixed at equal volumes to achieve a final concentration of lipids of 5 mg/mL and Jaspine B of 1 mg/mL, respectively, with the ethanol ratio kept constant.

Comments 3: Additionally, the authors should provide more details about the working solution.

Response 3: Thank you for your comment. We have provided that information under section 2.4. Preparation of working solutions and calibration standards on line 140.

Working solutions at concentrations of 0.5, 1, 2, 4, 8, and 16 ng/mL were prepared to construct a standard calibration curve of Jaspine B by serially diluting the stock solution (1 mg/mL) with methanol. Similarly, a 5 ng/mL working solution of Spisulosine, the internal standard (IS), was prepared by diluting the stock solution of 1 mg/mL Spisulosine in methanol. Calibration standard solutions were prepared by spiking blank rat plasma with 100 µL of Jaspine B and IS working solutions.

Comments 4: It would be helpful to see why the authors added the Jasmin dose as 5 m/kg.

Response 4: Thank you for the comment. We explained it as follows. Lines169-170

This dose was chosen to be consistent with our previous pharmacodynamic study [6].

Comments 5: The details of these abbreviation, TFF and TRF

The abbreviations TFF and TRF need to be fully explained, and more discussion is required, with additional supporting references.”

Response 5: Thank you for the comment. It has been addressed on lines 109-111.

Comments 6: More discussion and supported references are required.

Response 6: Thank you for the comments. Some extra references are added to relevant sections of the manuscript.

Comments 7: What about the degradation of liposomes after oral application.

Response 7: Thank you for your comments. The degradation aspect of the liposomal formulation after oral administration was not in the scope of this pilot study. We are hoping to investigate it in future studies. This issue has been mentioned in the study’s limitations. Lines 352-354.

Reviewer 2 Report (Previous Reviewer 4)

Comments and Suggestions for Authors

I have reviewed the authors’ responses and the revised manuscript. I appreciate their efforts to address the points raised during the initial review, including the addition of methodological explanations, clarification of the study’s limitations, and improvements to the manuscript’s presentation. However, despite these efforts, several critical concerns remain unresolved, which limit the overall scientific validity of the study. In particular, core issues related to the absence of TEM data, in vitro release studies, and biodistribution analysis have not been adequately addressed. These elements are essential to fully support the claims of enhanced bioavailability and sustained release mechanisms. Therefore, I recommend a Major Revision, with a strong suggestion that the authors either include additional experimental data or explicitly acknowledge the limitations and speculative nature of the mechanistic interpretations presented.

Lack of TEM Data

I acknowledge the authors’ explanation that the formulation method and optimization were based on a previous study. However, without TEM images or quantitative data in the current manuscript, it is impossible to verify the morphology, size uniformity, and integrity of the liposomes. As particle size and morphology are critical parameters that directly influence the pharmacokinetics and bioavailability of the formulation, the inclusion of representative TEM data is necessary to substantiate the study's claims.

If the authors do not intend to include TEM images, the mention of TEM analysis should be removed from both the Abstract and the Methods sections, as it creates confusion and raises concerns about the completeness of the experimental data.

Absence of In Vitro Release Data

The authors suggest that the liposomal formulation enhances bioavailability through potential controlled release effects. However, no in vitro release profile is presented to support this hypothesis. While it is understandable that the focus of the study is pharmacokinetics, the claim of controlled release remains speculative without supporting data. Including at least basic in vitro release studies would strengthen the mechanistic interpretation of the pharmacokinetic results.

Biodistribution Data Not Provided

The authors state that this is a pilot pharmacokinetic study and acknowledge the absence of biodistribution data as a limitation. While I appreciate this clarification, biodistribution data would have been valuable to support the claim that the liposomal formulation enhances absorption and prolongs circulation. Without such data, the mechanistic explanation for the improved pharmacokinetic profile remains incomplete.

Small Sample Size (N = 3)

The authors explain that the study is a pilot study, and I acknowledge the practical challenges involved. However, a sample size of N = 3 is generally insufficient for robust pharmacokinetic analysis, as it does not adequately account for inter-individual variability. While the pilot nature of the study is acceptable, the authors should clearly emphasize this limitation in both the Discussion and Conclusion sections.

Mechanistic Interpretation Remains Limited

The authors have added some discussion regarding possible mechanisms, such as enhanced absorption and prolonged circulation. However, these explanations are largely theoretical and not directly supported by experimental data. I encourage the authors to frame these interpretations cautiously and clearly acknowledge their speculative nature in the absence of direct evidence.

Author Response

Reviewer 2:

We appreciate your careful evaluation of our manuscript. Please find our detailed responses below. Revisions have been made accordingly and are indicated in the updated manuscript as highlighted text in relevant sections.

Additionally, the suggestion for language improvement has been addressed by our institution’s professional editing service.

Comments and Suggestions for Authors

Comments 1: I have reviewed the authors’ responses and the revised manuscript. I appreciate their efforts to address the points raised during the initial review, including the addition of methodological explanations, clarification of the study’s limitations, and improvements to the manuscript’s presentation. However, despite these efforts, several critical concerns remain unresolved, which limit the overall scientific validity of the study. In particular, core issues related to the absence of TEM data, in vitro release studies, and biodistribution analysis have not been adequately addressed. These elements are essential to fully support the claims of enhanced bioavailability and sustained release mechanisms. Therefore, I recommend a Major Revision, with a strong suggestion that the authors either include additional experimental data or explicitly acknowledge the limitations and speculative nature of the mechanistic interpretations presented.

Thank you for pointing this out. We agree with this comment. We have included additional data and explicitly acknowledged the limitations of our study regarding the other mentioned issues.

Comments 2: Lack of TEM Data

I acknowledge the authors’ explanation that the formulation method and optimization were based on a previous study. However, without TEM images or quantitative data in the current manuscript, it is impossible to verify the morphology, size uniformity, and integrity of the liposomes. As particle size and morphology are critical parameters that directly influence the pharmacokinetics and bioavailability of the formulation, the inclusion of representative TEM data is necessary to substantiate the study’s claims.

If the authors do not intend to include TEM images, the mention of TEM analysis should be removed from both the Abstract and the Methods sections, as it creates confusion and raises concerns about the completeness of the experimental data.

Response 2: We appreciate your comment. Although such data has been reported by our lab previously [6], a different version of those data has been added under section 2.2. Preparation and Characterization of Liposomal Formulation of Jaspine B…, line 99, and under result section 3.1. Liposome formulation and characterization, line 187 and Figure 1.

Comments 3: Absence of In Vitro Release Data

The authors suggest that the liposomal formulation enhances bioavailability through potential controlled release effects. However, no in vitro release profile is presented to support this hypothesis. While it is understandable that the focus of the study is pharmacokinetics, the claim of controlled release remains speculative without supporting data. Including at least basic in vitro release studies would strengthen the mechanistic interpretation of the pharmacokinetic results.

Response 3: Thank you for your valid comment and acknowledgment that the study’s focus was pharmacokinetics. We have designed this study to investigate the improved pharmacodynamic effects of Jaspine B after administration of its liposomal formulation compared to its plain suspension formulation in an animal model. We appreciate your comment on the necessity of in vitro studies to elucidate the possible mechanisms underlying the improvement of the Jaspine B parameters observed in this study and not firmly making a solid conclusion. Therefore, we revised our claim of controlled release in the Discussion section and mentioned our plan to continue collecting supporting data in future studies. 

Comments 4: Biodistribution Data Not Provided

The authors state that this is a pilot pharmacokinetic study and acknowledge the absence of biodistribution data as a limitation. While I appreciate this clarification, biodistribution data would have been valuable to support the claim that the liposomal formulation enhances absorption and prolongs circulation. Without such data, the mechanistic explanation for the improved pharmacokinetic profile remains incomplete.

Response 4: Thank you for your comment. Please see our explanation in response to the previous comment, as we have revised our discussion accordingly.

Comments 5: Small Sample Size (N = 3)

The authors explain that the study is a pilot study, and I acknowledge the practical challenges involved. However, a sample size of N = 3 is generally insufficient for robust pharmacokinetic analysis, as it does not adequately account for inter-individual variability. While the pilot nature of the study is acceptable, the authors should clearly emphasize this limitation in both the Discussion and Conclusion sections.

Response 5: Thank you for your comment. Your point is taken, and the limitation section has been revised accordingly!

Comments 6: Mechanistic Interpretation Remains Limited

The authors have added some discussion regarding possible mechanisms, such as enhanced absorption and prolonged circulation. However, these explanations are largely theoretical and not directly supported by experimental data. I encourage the authors to frame these interpretations cautiously and clearly acknowledge their speculative nature in the absence of direct evidence.

Response 6: Thank you for your comment. Your point is taken, and the limitation section has been revised accordingly!

Reviewer 3 Report (Previous Reviewer 5)

Comments and Suggestions for Authors

The current study aimed to enhance the oral bioavailability of Jaspine B through liposomal encapsulation:

1-Mention in a more detailed manner the role of liposomes in enhancing oral bioavailability of poorly soluble drugs in the introduction section.

2-The current study lacks several tests as ex vivo permeation studies, stability of liposomal formula in gastric and intestinal conditions, cytotoxicity study to assess the biocompatibility of formula.

3-Mention the encapsulation efficiency and zeta potential of the prepared formula.

Author Response

Reviewer 3:

We appreciate your careful evaluation of our manuscript. Please find our detailed responses below. Revisions have been made accordingly and are indicated in the updated manuscript as highlighted text in relevant sections.

Comments and Suggestions for Authors

The current study aimed to enhance the oral bioavailability of Jaspine B through liposomal encapsulation:

Comments 1: Mention in a more detailed manner the role of liposomes in enhancing oral bioavailability of poorly soluble drugs in the introduction section.

Response 1: Thank you for your comment. Your point is taken, and the requested information has been added as a full paragraph. Lines 53-60!

Comments 2: The current study lacks several tests as ex vivo permeation studies, stability of liposomal formula in gastric and intestinal conditions, cytotoxicity study to assess the biocompatibility of formula.

Response 2: Thank you for your valid comment. We designed this study to investigate the improved pharmacodynamic effects of Jaspine B after administration of its liposomal formulation compared to its plain suspension formulation in an animal model. The stability of the liposomal formula in gastric and intestinal conditions, as well as cytotoxicity, was not within the scope of this pilot study. We are hoping to investigate it in future studies. This issue has been mentioned in the study’s limitations. Lines 352-354.

Comments 3: Mention the encapsulation efficiency and zeta potential of the prepared formula.

Response 3: Thank you for your comment. The requested data has been added to section 3.1. Liposome formulation and characterization. Line 187!

Round 2

Reviewer 2 Report (Previous Reviewer 4)

Comments and Suggestions for Authors

Thank you for your response. I confirm that the manuscript has been appropriately revised according to the review comments.

Author Response

Thank you for agreeing that the manuscript has been appropriately revised.

Reviewer 3 Report (Previous Reviewer 5)

Comments and Suggestions for Authors

The authors have replied to the comments however the methodology of EE%was not mentioned in the revised version nor zeta potential results

Author Response

Thank you for accepting our response to your comments. We apologize that we missed the methodology for the EE% and zeta potential results. That information has been added to the manuscript.

Round 3

Reviewer 3 Report (Previous Reviewer 5)

Comments and Suggestions for Authors

The authors have performed the requested corrections 

This manuscript is a resubmission of an earlier submission. The following is a list of the peer review reports and author responses from that submission.

Round 1

Reviewer 1 Report

Comments and Suggestions for Authors
  1. The authors should expand the discussion by opposing nano-liposomes with other drug delivery systems (e.g., polymeric nanoparticles, micelles, or solid lipid nanoparticles) to set up the main advantages of the chosen formulation.
  2. The authors should provide more details for the rationale for choosing liposomal composition, e.g., how each component influences stability, drug loading, and release profile.
  3. While the pharmacokinetic advantage of Jaspine B is well established, the manuscript does not discuss whether these advantages could increase therapeutic activity in vivo.
  4. The n = 3 per group in the pharmacokinetic study is relatively low, which can affect statistical significance and reproducibility. A larger sample size could provide greater statistical analysis. If a larger sample size is not available, discuss this as a limitation in the manuscript.
  5. Stability data of the liposomal product (e.g., physical stability, drug loss with time) are not.
  6. Small language mistakes must be removed by proper proofreading.
  7. Some legends lack sufficient details about the experimental conditions.
  8. The limitations of the study should be included.

Reviewer 2 Report

Comments and Suggestions for Authors

The manuscript presents a well-structured study on the pharmacokinetics of Jaspine B using a nano-liposomal drug delivery system. Before publication, the authors should consider the following points:

  1. The introduction is informative. However, the information on Jaspine B’s poor oral bioavailability should be summarized efficiently. Further, comparative information on other bioavailability enhancement strategies (e.g., solid lipid nanoparticles) could provide additional context.
  2. The pharmacokinetic data is well-presented, but authors are expected to provide with a more detailed statistical comparison between liposomal and free Jaspine B.

  3. The mechanism behind improved absorption and prolonged circulation should be explored in more depth in the results and discussion section. By the way, there is no section 4 before "5. conclusion".

Reviewer 3 Report

Comments and Suggestions for Authors

While the study presents an interesting approach to improving the bioavailability of Jaspine B, it suffers from several significant flaws, including a lack of novelty, insufficient pharmacodynamic data, a small sample size, and incomplete methodological details. These issues collectively adversely affect the study's validity and impact. Therefore, I recommend rejecting the article in its current form.

Reviewer 4 Report

Comments and Suggestions for Authors

 This study aimed to improve the low bioavailability of Jaspine B by developing a liposomal formulation and evaluating its pharmacokinetics in a Sprague Dawley rat model. The researchers used LC-MS/MS to measure plasma drug concentrations and reported that the liposomal formulation increased the half-life (t1/2), total exposure (AUC), and mean residence time (MRT). Jaspine B is a highly potent anticancer compound, but previous studies have reported an extremely low oral bioavailability of 6.2%. Therefore, efforts to enhance its bioavailability are scientifically valuable. Additionally, developing a liposomal formulation using microfluidic techniques is an innovative approach, and applying LC-MS/MS for quantitative analysis is methodologically appropriate. However, the study has several limitations, including weaknesses in experimental design, insufficient methodological explanations, and unclear interpretation of results, all of which affect the overall reliability of the findings. In particular, issues such as the small sample size (N=3) in the pharmacokinetic study, limited characterization of the liposomal formulation, and the lack of a mechanistic explanation for the shortened Tmax alongside an increased half-life need to be addressed. Given these limitations, the study lacks sufficient reliability and reproducibility in its current form. Therefore, additional experiments and further revisions are necessary. I recommend rejection of this manuscript.

Major Issues and Areas for Improvement

  1. Reliability Concerns Due to Small Sample Size

In pharmacokinetic studies, a sample size of N=3 is insufficient to account for inter-individual variability. To ensure the reliability of the experimental results, a minimum of N = 6~8 samples should be included. With the current data, it is difficult to confirm reproducibility.

  1. Lack of Detailed Description of Liposome Preparation and TEM Data

Although the abstract mentions the use of microfluidic techniques, the main text does not provide a detailed explanation of the liposome preparation process. Additionally, the manuscript mentions TEM analysis, but no TEM data or results are presented, making it unclear whether this experiment was actually conducted.

  1. Insufficient Explanation of the Liposomal Drug Delivery Mechanism

The manuscript reports that the liposomal formulation improved bioavailability, but it does not adequately explain the mechanism behind this enhancement. Further discussion is needed on possible mechanisms such as enhanced intestinal absorption, lymphatic uptake, and controlled drug release.

  1. Lack of Clear Interpretation of Experimental Results

The manuscript does not provide a sufficient explanation for the simultaneous decrease in Tmax and increase in half-life (t1/2). Typically, faster absorption (shorter Tmax) is associated with a shorter half-life, yet the study reports the opposite trend. Although the manuscript includes plasma concentration-time profiles, it does not offer a mechanistic interpretation of these pharmacokinetic changes. If the liposomal formulation is modulating drug release, additional studies such as drug release rate analysis and tissue distribution studies would be necessary to support this claim.

  1. Insufficient Discussion on Clinical Applications

The manuscript states that the liposomal formulation enhances bioavailability, but it lacks discussion on its potential clinical benefits, such as whether it could reduce the required dosage or minimize side effects. Since Jaspine B has potential as an anticancer agent, further discussion on its clinical applicability and future research directions would strengthen the manuscript.

Comments on the Quality of English Language

My english proficiency is not sufficient to assses the quality of the manuscript.

Reviewer 5 Report

Comments and Suggestions for Authors

The current study aimed to formulate Jaspine B-loaded liposomal formula for enhanced bioavailability; the following should be addressed:

1- The abstract contains no numerical findings.

2- Please add a reference for the animal dose of  Jaspine B.

3- The authors stated in the abstract that Jaspine B liposomes were prepared using microfluidic techniques and characterized using transmission electron microscopy (TEM). However, the preparation is not mentioned in the methodology section, nor is the result of the TEM in the result section.

4-Please provide a brief summary of the preparation of liposomal formulae and their characteristics, such as particle size, PDI, and surface charge.

4- Please mention the software used to estimate PK parameters.

5- In the introduction section, provide a brief regarding the previous studies that utilized pegylated liposomes for enhancing PK parameters of poorly soluble drugs.

6-A Biodistribution studies in various organs should be better performed to emphasize the difference between drug suspension and liposomal formula.

7- The weight of rats should be mentioned.

8- The time unit is h, not hr.